# The impact of digital economy on the upgrading of manufacturing structure

**Ting Chen**[1]*, **Songlan Zhou**[2]

**1** College of Digital economy and Trade, Guangzhou Huashang University, Zengcheng District, Guangzhou City, Guangdong Province, China, **2** School of Economics and Management, Guangzhou College of Applied Science and Technology, Guangzhou City, Guangdong Province, China

* 2652570568@qq.com

## Abstract

The global economic situation is in a downturn, and the upgrading of manufacturing structure is a necessary transformation path for the manufacturing industry to achieve high-speed and stable development. The article analyzes the theoretical mechanism of the digital economy affecting the upgrading of manufacturing structure through the endogenous economic growth model, constructs a three-stage mediation effect model, and empirically researches the path of the digital economy affecting the upgrading of manufacturing structure in the Pearl River Delta. The study finds that the digital economy has a significant positive promoting effect on the upgrading of manufacturing structure. In terms of the influencing mechanism, the enhancement of the level of independent innovation and the advancement of the human capital structure are the important paths of the digital economy in promoting the upgrading of manufacturing structure. Among them, the mediating effect is 17.5% for the level of independent innovation and 17.4% for the level of the advancement of the human capital structure. The results of the study also found that the upgrading of manufacturing structure cannot be separated from government support, and the influence of government support on t the upgrading of manufacturing structure reaches 44.9%, and government deployment and control is conducive to accelerating the process of advanced manufacturing structure.

**Data Availability Statement:** The data used in this research are third party data collected by myself, and anyone can legally obtain these data through the China Statistical Yearbook (https://www.stats.gov.cn/sj/ndsj/) and the Guangdong Statistical Yearbook (https://tjj.gz.gov.cn/datav/admin/home/

## Introduction

In recent years, the COVID-19 has broken out, and the uncertainty of the international situation has increased. China's manufacturing industry is facing dual pressures from home and abroad. For external reasons, trade frictions and trade protection have intensified. And there is still a large gap between the level of high-tech and that of developed countries. For internal reasons, the demographic dividends have gradually disappeared, supply side structural reform and core technology constraints. Those all restrict China's transformation from a "big country" to a "strong country" in manufacturing. Under these factors, the proportion of added value of China's manufacturing industry to GDP has continued to decline. Compared to 2010, the proportion of added value of China's manufacturing industry to GDP dropped continuously from

www_nj/). The Supporting Information file S1 is the data result processed by myself based on the collected data.

**Funding:** This work was supported by Guangzhou Huashang University Intramural Research Mentorship Program Grant:" The impact of the digital economy on the structural upgrading of manufacturing" (NO. 2024HSDS05), Ting Chen is the project leader and project facilitator. This work was also supported by the Natural Science Foundation of China: "Gap Measurement, Leading Mechanism, and Innovation Leap Research of New Science and Technology Revolution Pilot Technology" (No. 71974041), Songlan Zhou is the project leader and project facilitator. The funders had no role in study design, data collection and analysis, decision to publish, or preparation of the manuscript.

**Competing interests:** The authors have declared that no competing interests exist.

about 32.46% to 26.29% in 2020. Compared with the previous year, the proportion of added value of China's manufacturing industry to GDP increased slightly, accounting for 27.55% in 2021, and the growth rate of added value of China's manufacturing industry was about 18.83% (The data comes from China National Statistical Yearbook from 2014 to 2023). In the "Made in China 2025" issued by the State Council in 2015, it was pointed out that China's manufacturing industry is still large but not strong, and there is a significant gap between China and a manufacturing powerhouse in terms of independent innovation ability, resource utilization efficiency, industrial structure level, informatization level, etc. The upgrading of the manufacturing industry structure is urgent. In this context, the digital economy, with its high integration and strong penetration, permeates various industries and is accelerating the transformation of traditional industries to digital industries in China.

The digital economy, with digital technology as its core, provides a core driving force for the advancement of manufacturing structure. The 20th National Congress of the Communist Party of China clearly pointed out that China should accelerate the development of the digital economy, strengthen the integration of the digital economy and industry, and create a digital industry with international competitiveness. Vigorously developing an advanced manufacturing system, including high-tech manufacturing and high-end equipment manufacturing, is the main direction for upgrading the industrial structure of China's manufacturing industry, and it has great development potential and space [1]. Therefore, promoting the integration of manufacturing and digital economy, promoting the upgrading of manufacturing structure, is of great significance for the high-quality development of manufacturing.

## Literature review

### The upgrading of manufacturing structure

Scholars mainly focus on industrial integration, human capital, environmental regulation, government support and technological progress to study the relationship with the upgrading of manufacturing structure. Xue-Jun L et al (2016) [2] argued that the rise of informatization industry and Internet+ provides a new path for industrial structure upgrading, and the integration of information technology and manufacturing industry is conducive to the upgrading of manufacturing structure. Chengkun Liu (2021) [3] tested through the spatial effect that the quality of human capital is positively correlated with the upgrading of manufacturing structure, and human capital has a positive impact on industrial structure upgrading [4]. Porter et al. (1995) [5] hypothesis that, although environmental regulations in the short term will lead to an increase in enterprise costs, but in the long term, enterprises will avoid the negative impact of environmental regulations by strengthening technological innovation and management, which indirectly promotes the upgrading of industrial structure. Xiqiang Chen and Yuanhai Fu (2017) [6] indicate that the government tends to form an administrative monopoly, industrial policy distortion, local protection, market segmentation and so on, leading to resource mismatch, is not conducive to the upgrading of manufacturing structure. However, Xiangsong Ye and Jing Liu (2020) [7] believes that government support strongly promotes the progress of high-end manufacturing science and technology level, and technological progress is conducive to the upgrading of manufacturing structure [8]. Jingrong Dong and Wenqing Zhang (2019) [9]classified the sources of technological progress into four kinds of technological imports, foreign investment, cooperative research and development (R & D), and independent R & D, and estimated them through the method of SUR, analyzing their relationship with the upgrading of manufacturing structure, and the study found that technology import, cooperative R&D as the source of technology has a significant positive promotion effect on the upgrading of manufacturing structure in China, but for the technology-intensive eastern

region, the upgrading of manufacturing structure is mainly affected by foreign direct investment, independent R&D.

## Digital economy and the upgrading of manufacturing structure

The digital economy represented by Big data, artificial intelligence, Internet plus and information technology has realized the deep integration of digital technology and the real economy, promoting the accelerated transformation of traditional manufacturing industry. From the perspective of factor allocation, the digital economy realizes information transparency through Big data, reduces manufacturing production costs, optimizes resource allocation, drives innovative development, improves manufacturing enterprise production efficiency [10], and further promotes the upgrading of manufacturing structure. From the perspective of impact approach, Yong Zhou et al. (2022) [11] found that the digital economy has significantly promoted the upgrading of manufacturing structure through the study of the mediating effect model, in which innovation ability and Total factor productivity have a significant partial mediating effect on this promotion. In addition, Yanze Cai et al. (2021) [12]pointed out that the innovation environment, including talent gathering and financial development, plays a moderating role in the digital economy's significant promotion of the upgrading of manufacturing structure, And this regulatory effect has a certain threshold effect. When talent aggregation and financial development reach a certain level, the effect of digital economy on promoting the upgrading of manufacturing structure is significantly enhanced.

Based on existing research, this study has two innovations, first, by using an endogenous economic growth model, the impact of the digital economy on the upgrading of manufacturing structure is analyzed, providing theoretical support for the impact of the digital economy on industrial growth rate. Existing research mostly focuses on empirical analysis of the impact of various economic variables on the upgrading of manufacturing industry structure, but generally lacks an inherent theoretical analysis of the impact of the digital economy on the upgrading of manufacturing industry structure. This article takes into account the characteristics of the digital economy and describes it as "knowledge", which can be a technology that affects labor efficiency or a theory that affects capital operation efficiency. It integrates the Cobb Douglas production function and explores the internal mechanism of the impact of the digital economy on the upgrading of manufacturing structure. Second, Considering the transmission effect of economic variables and the diffusion effect of digital economy, this article uses human capital and technological progress as intermediary variables to analyze the impact path of digital economy on the upgrading of manufacturing structure. On the other hand, the article focuses on the Pearl River Delta, as a frontier area of advanced technology, capital accumulation and economic development, it has a strong reference significance on whether and how the digital economy affects the upgrading of manufacturing structure. It is of great significance to promote the integration of digital economy and manufacturing industry, accelerate the transformation of traditional manufacturing industry in the Pearl River Delta, and break through the bottleneck of manufacturing industry development in the Pearl River Delta.

## Research hypothesis

With the advent of the Internet and big data, data elements, digital technology, and digital economy have emerged. The digital economy, with information and communication technology (ICT) as its core, is the third new economic form brought about by informatization, following agricultural and industrial economies. According to the endogenous economic growth model, technological progress is endogenous, and capital is divided into physical capital and knowledge capital. The former has the characteristic of diminishing returns to scale, while the

latter is not. Knowledge investment is the key to high equilibrium growth rates. Knowledge has non exclusivity, and both the capital and human capital departments can use the entire knowledge stock (A). In the endogenous economic growth model, product A produced by research and development is described as knowledge, which exists in many forms, including technologies that improve labor efficiency and theories that improve capital operation efficiency. This characteristic is completely consistent with the characteristics of the digital economy. The digital economy can not only optimize management models through digital technology [13], improve the efficiency of production, exchange, distribution, and consumption, reduce transaction costs, and promote enterprises to achieve economies of scale [14] but also the data elements derived from the digital economy play a central role in production, it improves the synergy between labor, capital, and other factors by utilizing valuable information [15]. Data elements penetrate through various stages of production through high penetration rates, accelerating capital operation, greatly shortening the capital operation cycle, and doubling the efficiency of capital operation. Therefore, this article believes that the digital economy can affect both labor and capital, and lists the following Cobb Douglas production function:

$$Y(t) = [(1 - a_k)A(t)K(t)]^{\alpha}[(1 - a_L)A(t)L(t)]^{1-\alpha}, \ \ 0 < \alpha < 1 \tag{1}$$

Calculate:

$$Y(t) = A(t)[(1 - a_k)K(t)]^{\alpha}[(1 - a_L)L(t)]^{1-\alpha}, \ \ \ \ 0 < \alpha < 1 \tag{2}$$

Neglecting depreciation of capital, the change in capital is

$$\dot{K}(t) = sY(t) = s(1 - a_k)^{\alpha}(1 - a_L)^{1-\alpha}A(t)K(t)^{\alpha}L(t)^{1-\alpha}, \ \ 0 < \alpha < 1 \tag{3}$$

$$\text{If } c = s(1 - a_k)^{\alpha}(1 - a_L)^{1-\alpha}$$

$$g_{k=}\frac{\dot{K(t)}}{K(t)} = cA(t)\left[\frac{L(t)}{K(t)}\right]^{1-\alpha} \tag{4}$$

Taking logarithmic derivative over time yields:

$$\frac{\dot{g_k}}{g_k} = g_A + (1 - \alpha)(n - g_k) \tag{5}$$

$$\text{When } \dot{g_k} = 0$$

$$g_k = \frac{g_A}{1 - \alpha} + n \tag{6}$$

Y, K and L respectively represent output, capital, and labor, $a_k$ is the proportion of capital invested in research and development, and $(1 - a_k)$ is the proportion of capital invested in production. $a_L$ is the proportion of research and development investment in labor. $(1 - a_L)$ is the proportion of labor input into production. $g_A$ is the growth rate of knowledge. $g_k$ is the growth rate of capital, and n is the growth rate of labor. The production of new knowledge depends on

the capital, labor, and technological level of research, thus obtaining the following equation:

$$A\dot{(}t) = \mathrm{B}[a_k\mathrm{K}(t)]^{\beta}[a_L L(t)]^{\gamma}\mathrm{A}(t)^{\theta} \quad ,\mathrm{B} > 0, \beta \geq 0, \gamma \geq 0 \tag{7}$$

$$g_A = \frac{A\dot{(}t)}{A(t)} = \mathrm{B}[a_k\mathrm{K}(t)]^{\beta}[a_L L(t)]^{\gamma}\mathrm{A}(t)^{\theta-1} \tag{8}$$

Taking logarithmic derivative over time yields:

$$\frac{\dot{g_A}}{g_A} = (\theta - 1)g_A + \beta g_k + \gamma n \tag{9}$$

When $\dot{g_A} = 0$

$$g_k = \frac{1-\theta}{\beta}g_A - \frac{\gamma n}{\beta} \tag{10}$$

B is the conversion parameter, θ is the impact of knowledge stock on R&D rate. If θ is greater than 1, it indicates that the knowledge stock has a huge impact on the production of new knowledge. The marginal increase in the level of knowledge stock will generate a large amount of new knowledge, leading to a continuous increase in the growth rate of knowledge. If θ Equal to 1, the increase in knowledge stock will be proportional to the addition of new knowledge. If θ is less than 1, it indicates that the increase in knowledge stock has a limited effect on new knowledge and will gradually converge. According to the characteristics of the digital economy, as a new production factor, the digital economy integrates with various industries, eliminates industrial barriers, avoids adverse selection and external diseconomy caused by information asymmetry in industries, effectively saves enterprise operating costs and menu costs, improves industrial allocation efficiency, and forms Pareto optimality. Those have a huge impact on the increase of new knowledge. Therefore, this article assumes that in the context of the digital economy, θ> 1. Because β≥ 0, θ+ β> 1. Draw the following coordinate graph according to Eqs (6) and (10), as shown in Fig 1, with the two curves gradually separating.

The initial values of model parameters and knowledge (A), capital (K), and labor (L) determine the initial value of $g_A$, $g_k$. According to Fig 1, in the context of the digital economy, no matter where the initial value of $g_A$, $g_k$ is, it will enter the middle region between the curve $\dot{g_k} = 0$ and $\dot{g_A} = 0$, and $\dot{g_k}$, $\dot{g_A}$ will be greater than 0. In other words, $g_k$, $g_A$ will continue to grow, and the growth rate of capital and knowledge will continue to increase. According to the production function Eq (2), the output growth rate can be written as:

$$\frac{\dot{Y}(t)}{Y(t)} = g_Y = g_A + \alpha g_k + (1 - \alpha)n \tag{11}$$

According to Eq (11) and the analysis above, in the context of the digital economy, $g_A$ and $g_k$ are always greater than 0, and $\alpha$ is greater than 0 and less than 1. Therefore, the left side of the equation is always greater than 0, as the growth rate of knowledge and capital continues to increase, the output growth rate also continues to increase. What's more, the Kuznets rule points out that within the manufacturing industry, the fastest-growing sectors are emerging industries closely related to modern technology. Based on the analysis above, this article believes that the digital economy can not only comprehensively promote the increase of

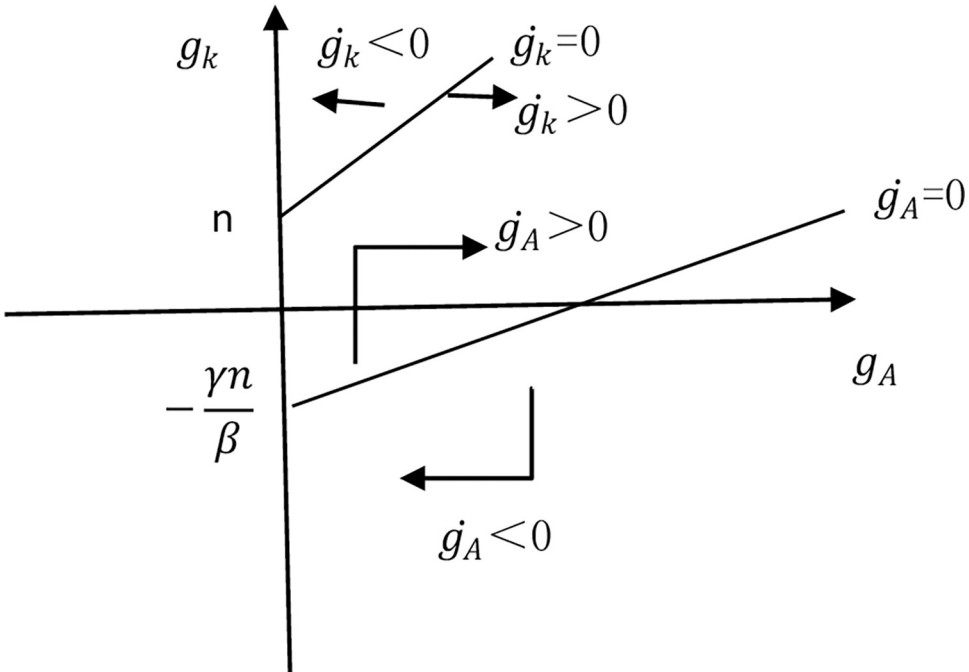

**Fig 1. Capital and knowledge growth rate curve.**

manufacturing output, but also have a more significant promoting effect on high-end technology sectors. Therefore, the following hypothesis is proposed.

Hypothesis 1: The digital economy can promote the upgrading of manufacturing structure.

Olena Oliinyk (2021) [16] states that factors such as the ability of new technologies to work with people, the ability to innovate, and the means of communication become determinants of the efficiency of economic development, the shortage of skilled workers slows down the development of business and leads to additional costs for the development of human capital [17], information and communication technologies are able to drive economic growth [18], and increasing the efficiency of innovation is essential for creating competitive advantages [19]. Accelerating the development of the digital economy helps to promote technological progress and the accumulation of human capital, thereby assisting in the structural adjustment of the manufacturing industry, which is of great significance for promoting high-quality development of the manufacturing industry.

The development of the digital economy has accelerated the progress of industrial digitization and digital industry, improved capital allocation and utilization efficiency, provided sufficient funds for R & D innovation, stimulated innovation vitality, and thus promoted technical progress [20]. From the supply side perspective, technical progress injects new momentum into the manufacturing industry dominated by information technology, improves enterprise production efficiency, changes traditional production methods, reduces production costs, increases producer surplus, and promotes the upgrading of manufacturing structure. From the demand side, technical progress can create more diverse and diverse goods for consumers, provide more convenient and efficient services, enhance consumer experience, and drive consumption. The increase in consumer demand will inevitably promote technical progress, thereby promoting the upgrading of manufacturing structure. The new economic growth theory points out that technical progress is conducive to improving the core competitiveness of

industries and achieving a leap from low to high added value in industries [21]. Therefore, this article proposes a second hypothesis:

Hypothesis 2: The digital economy promotes the upgrading of manufacturing structure by promoting technology progress

The digital economy is the product of the information age. With the development of digital technologies such as artificial intelligence and Big data, traditional human capital can no longer meet the requirements of digital technology. The improvement of digital technology cannot be separated from the accumulation of scientific knowledge and technological innovation ability of high digital literacy talents [22]. On the one hand, in order to enhance their own development space, win better working conditions, and obtain more job opportunities, workers continuously improve their professional knowledge and skills through education and training to meet the needs of the times [23]. The digital economy has promoted the advancement of human capital structure through the effects of expansion, deepening, and career creation [24]. On the other hand, according to the theory of human capital, the accumulation and upgrading of human capital structure are the third fundamental change that occurs in productivity. Human capital has the increasing effect of returns to scale, which is conducive to the increase of income from other input factors. And it is the basis of industrial structure change, and also determines the direction, speed and effect of industrial structure change. Human capital structure plays an important role in industrial structure transformation by influencing production efficiency, innovation performance, and agglomeration effect [25]. Therefore, this article proposes a third hypothesis:

Hypothesis 3: The digital economy promotes the advancement of manufacturing structure through the advancement of human capital structure.

Based on the analysis above, the relationship between the digital economy and the upgrading of manufacturing structure is shown in Fig 2.

## Method and data

### Benchmark model

Based on the mechanism analysis of the impact of the digital economy on the upgrading of manufacturing structure mentioned above, in order to test the research hypothesis, the following benchmark regression model is constructed for the direct transmission mechanism of the

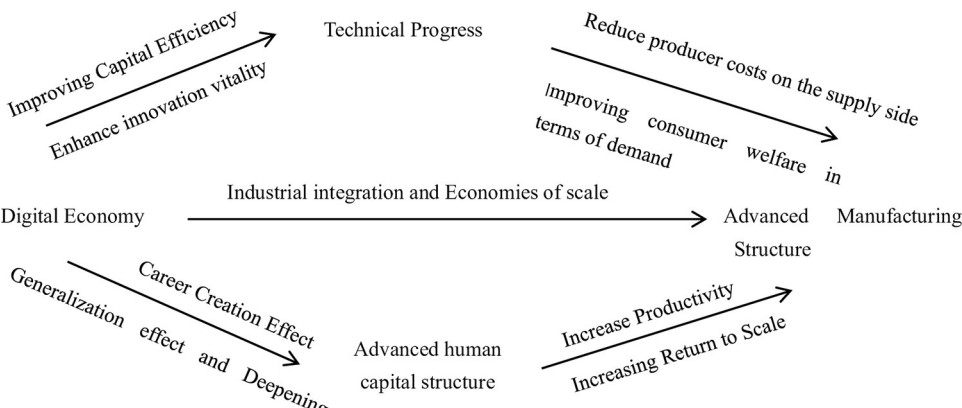

**Fig 2. The transmission path of the impact of digital economy on the upgrading of manufacturing structure.**

impact of the digital economy on the upgrading of manufacturing structure:

$$\text{ManuH}_{it} = \alpha_0 + \alpha_1 Digital_{it} + \sum\nolimits_k \alpha_2 Control + \delta_i + \varepsilon_i + \mu_{it} \tag{12}$$

Among them, i and t represent the sample individuals and time, respectively. ManuH represents the upgrading of manufacturing structure. Digital is the level of digital economy development calculated based on principal component analysis method. Control is the control variable. $\delta_i$ and $\varepsilon_i$ represents individual and time effects, respectively, $\mu_{it}$ is a random interference term.

## Mediated effect model

In addition to the direct effect reflected in Eq (12) above, in order to discuss the possible transmission mechanism of the impact of the digital economy on the upgrading of manufacturing structure, we test whether independent innovation, the import of technology, and human capital are intermediary variables between the two. The specific testing steps are as follows: On the basis of the significant passing of the coefficients in the linear regression model (12) of the digital economy on the upgrading of manufacturing structure, and then construct linear regression models (13), of the digital economy on intermediary variables and regression Eq (14) of the impact of the digital economy and intermediary variable on the upgrading of manufacturing structure:

$$\text{ME}_{it} = \beta_0 + \beta_1 \text{Digital}_{it} + \sum\nolimits_k \beta_2 \text{Control} + \delta_i + \varepsilon_i + \mu_{it} \tag{13}$$

$$\text{ManuH}_{it} = \gamma_0 + \gamma_1 \text{Digital}_{it} + \gamma_2 \text{ME}_{it} + \sum\nolimits_k \gamma_3 \text{Control} + \delta_i + \varepsilon_i + \mu_{it} \tag{14}$$

ME is mediating variable. According to the mediated effect model analysis steps of Zhonglin Wen and Lei Zhang (2004) [26], judge whether mediating effect exists and the type of mediating effect. The judgment steps are shown in Fig 3.

## Data sources

Considering the availability of data, this article uses data from nine cities in the Pearl River Delta from 2012 to 2021, all of which are sourced from the Guangdong Statistical Yearbook.

**Dependent variable.** The industrial structure upgrading of the manufacturing industry mainly involves the transformation and upgrading from resource and labor-intensive manufacturing to technology and capital intensive manufacturing, and from traditional low-end manufacturing to modern, advanced, and emerging manufacturing [1]. This article refers to the OECD's classification method for manufacturing industry and divides it into low-end manufacturing, mid-end manufacturing, and high-end manufacturing, among those, low-end manufacturing including food processing and manufacturing, beverages, tobacco, textiles, clothing, leather, wood, furniture, paper making, printing and sports goods, and other manufacturing industries. The middle-end manufacturing including petroleum processing, coking and nuclear pigment processing, rubber and plastics, non-metallic minerals, Ferrous smelting, non-ferrous metal smelting and metal products. And High-end manufacturing: chemical medicine, general equipment, specialized equipment, transportation equipment, electrical machinery and equipment, computer communication and electronic equipment, instruments and meters, etc. And the study refer to the approach of Zhanxiang, F et al.(2016) [27], adopt the proportion of high-end manufacturing output value to the total manufacturing output value to measure the degree of the upgrading of manufacturing structure. The larger the proportion, the higher the degree of the upgrading of manufacturing structure.

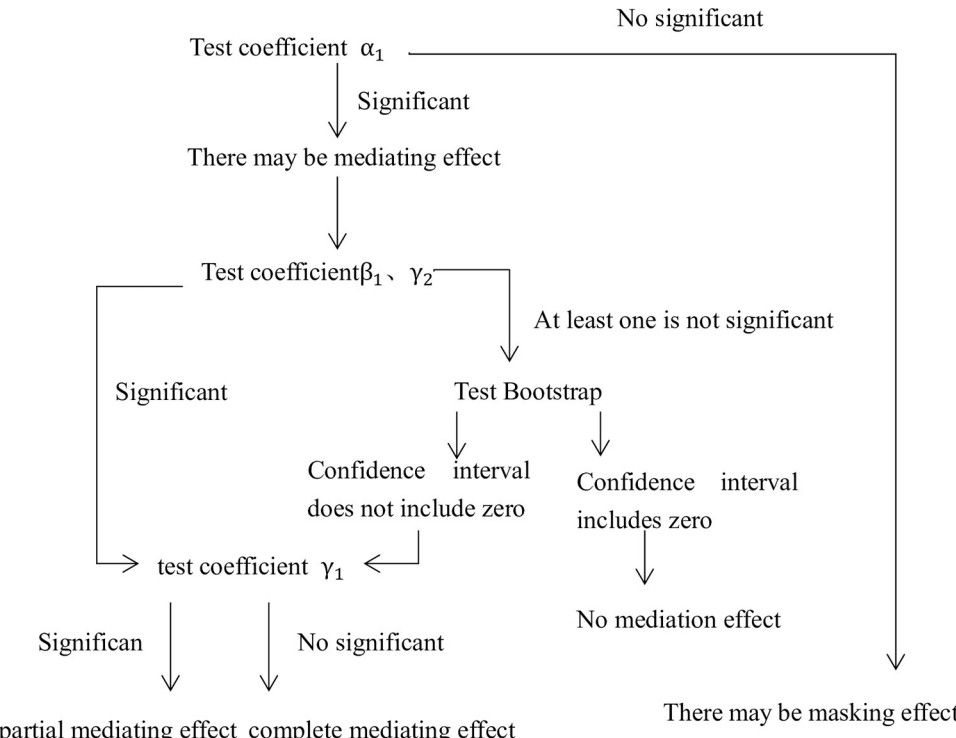

**Fig 3. Diagram of mediating effect.**

**Independent variable.** This article, from the perspective of digital application and output, selects five indicators, including internet penetration rate, mobile phone penetration rate, fixed line penetration rate, digital output, and digital technology related practitioners, and uses principal component analysis to synthesize a digital economy development index to illustrate the level of regional digital economy development.

**Mediating variable.** Technical progress is the fundamental way to optimize the structure of the manufacturing industry [28]. The article divides technical progress into two aspects: independent innovation and the import of technology. Scholars mainly measure independent innovation in terms of income and expenditure. This article refers to Jie Zhang et al. (2020) [29], who use per capita scientific and technological activity expenditure to measure the level of independent innovation in terms of expenditure. Considering the impact of the import of technology on domestic enterprises, foreign investment participation is adopted that the proportion of total output value of foreign enterprises to total industrial output value to measure the level of the import of technology.

Advanced human capital structure: The level of human capital can improve the labor efficiency of the manufacturing industry and promote the upgrading of manufacturing structure. This article considers that the digital economy requires a high level of talent literacy, and the number of ordinary undergraduate graduates has not yet reflected the human capital level of digital technology talents. Therefore, this article uses the number of scientific and technological personnel to measure the advancement of human capital structure.

**Control variable.** This article selects the per capita GDP level to measure the economic development level of a region. Using the proportion of non-state-owned industrial output value to total industrial output value to measure the degree of marketization. The proportion of fiscal expenditure to GDP indicates government intervention. The ratio of total import and

export volume to GDP measures the level of opening-up. The ratio of year-end urban population to year-end permanent population measures the urbanization rate. The variable description and descriptive statistics are shown in Table 1.

## Empirical results

### Baseline analysis

This article uses a stepwise regression method to analyze the relationship between the digital economy and the upgrading of manufacturing structure by sequentially adding government intervention, urbanization, marketization and Level of opening-up in model (12), Table 2. According to the benchmark regression results in Table 2, the research results strongly indicate that the digital economy has a significant positive impact on the upgrading of manufacturing structure. For every 1 unit increase in the development level of the digital economy, the upgrading of manufacturing structure increases by 0.0227 units. The digital economy, through its highly technological characteristics, is conducive to stimulating the vitality of regional independent innovation, improving the level of independent innovation, promoting the development of industrial technology and the integration of technology and industry, promoting the generation of economies of scale and scope in industries, greatly reducing production costs, improving industrial efficiency, and promoting the upgrading of manufacturing structure, confirming the hypothesis 1.

According to (2) to (6) in Table 2, the impact of control variables on the upgrading of manufacturing structure is analyzed. The results show that government intervention is conducive to the upgrading of manufacturing structure in the Pearl River Delta region, and the impact is very significant. For every unit increase in government intervention, the upgrading of manufacturing structure increases by 0.45 units. In addition, the improvement of urbanization level and the degree of marketization are not conducive to the upgrading of manufacturing structure in the Pearl River Delta region. Due to the fact that urbanization is the result of population migration under factors such as production and consumption structure, income situation, and government expenditure distribution [30], urbanization can promote the transformation of employment structure and promote urban industrialization. Although urbanization can bring more labor and investment, it mainly targets low-end industries and has a significant crowding out effect on high-tech industries [31], which is not conducive to the upgrading of manufacturing structure. Furthermore, the improvement of marketization degree means that the government's control over the market economy is gradually relaxed,

**Table 1. Variable indicators and descriptive analysis.**

| Variable Type | Variables | Variable Symbols | Mean | Std. Dev. | Min | Max |
|---|---|---|---|---|---|---|
| **Dependent variables** | the upgrading of manufacturing structure | ManuH | 0.596 | 0.165 | 0.261 | 0.848 |
| **Independent variable** | Digital economy | Digital | 1.67e-08 | 0.977 | -1.825 | 2.544 |
| **Mediating variable** | The import of technology | Tech-int | 0.458 | 0.116 | 0.216 | 0.713 |
| | Independent innovation | Tech-inv | 94.131 | 178.173 | 2.111 | 769.221 |
| | Advanced human capital structure | High-hum | 2 123.99 | 4 705.67 | 58 | 191 94 |
| **Control variable** | Economic development level | PGDP | 9.018 | 3.594 | 3.240 | 17.366 |
| | Marketization | Mark | 0.861 | 0.097 | 0.556 | 0.959 |
| | Government intervention | Gov | 0.125 | 0.039 | 0.018 | 0.203 |
| | Level of opening-up | Open | 0.864 | 0.517 | 0.153 | 2.284 |
| | Urbanization | Urban | 0.805 | 0.164 | 0.426 | 1 |

**Table 2. Regression results.**

| Variables | (1) | (2) | (3) | (4) | (5) | Robustness test results (6) | (7) |
|---|---|---|---|---|---|---|---|
| | ManuH | ManuH | ManuH | ManuH | ManuH | ManuH | Manuhigh |
| Digital | 0.018 8*** (4.06) | 0.015 1*** (3.20) | 0.022 7*** (4.16) | 0.023 0*** (4.42) | 0.022 7*** (3.45) | | 0.02 299*** (3.09) |
| L.digital | | | | | | 0.022 9* 2** (2.94) | |
| Gov | | 0.296 8*** (2.46) | 0.367 3*** (3.06) | 0.455 4*** (3.85) | 0.449 4*** (3.16) | 0.339 3 2** (2.37) | 0.630 1*** (3.93) |
| Urban | | | -0.456 2** (-2.55) | -0.492 8*** (-2.88) | -0.490 7*** (-2.81) | -0.530 5 *** (-2.80) | -0.441 42** (-2.24) |
| Mark | | | | -0.175 8*** (-2.93) | -0.177 8*** (-2.70) | -0.218 6 *** (-3.44) | -0.350 9*** (-4.72) |
| Open | | | | | -0.001 2 (0.939) | -0.006 4 (-0.38) | -0.002 5 (-0.14) |
| Constant | 0.596 0*** (242.50) | 0.558 9*** (36.56) | 0.917 3*** (6.49) | 1.087 1*** (7.41) | 1.089 0*** (7.28) | 1.179 4 *** (7.03) | 1.079 3*** (6.39) |
| Observations | 90 | 90 | 90 | 90 | 90 | 81 | 90 |
| Fixed effect | Yes | Yes | Yes | Yes | Yes | Yes | Yes |
| R-square | 0.622 2 | 0.423 4 | 0.300 7 | 0.360 5 | 0.360 5 | 0.317 0 | 0.448 5 |

Notes

***, **, and * in the table denote significance levels at 1%, 5%, and 10%, respectively, and standard deviations are in parentheses, as below.

and the economy is automatically regulated by the market. Analysis shows that the upgrading of manufacturing structure in the Pearl River Delta region is undergoing a transformation with the development of the digital economy. The digital economy in the Pearl River Delta region is still in a stage of rapid development but the development level is not yet fully mature and uneven. The technological R & D, capital investment, and human development required for the upgrading of manufacturing structure still require strong government support. The improvement of marketization has a significant inhibitory effect on the technological level of high-end manufacturing industry, which is not conducive to the upgrading of manufacturing structure in the Pearl River Delta at present. According to the benchmark regression results in Table 2, it can also be found that level of opening-up has a negative impact on the upgrading of manufacturing structure in the Pearl River Delta, but this effect is not significant. Therefore, it can be seen that the increase in the degree of opening up of the Pearl River Delta to the outside world is not conducive to the upgrading of manufacturing structure in the Pearl River Delta.

## The analysis of mediating effect model

According to (8), (10), and (12) in Table 3, it shows that the digital economy has a significant impact on the level of independent innovation, the import of technology, and the upgrading of human capital structure. Among them, the development of the digital economy has a significant positive impact on the level of independent innovation and the upgrading of human capital structure. Due to the high-tech nature of the digital economy, the development of the digital economy provides broader development space and innovative vitality for independent innovation, thereby promoting the improvement of the level of independent innovation. The development of the digital economy cannot be separated from the demand for high-skilled

**Table 3. The analysis results of mediating effect model.**

| Variables | (8) | (9) | (10) | (11) | (12) | (13) |
|---|---|---|---|---|---|---|
| | Tech-inv | ManuH | Tech-int | ManuH | Highhum | ManuH |
| Digital | 69.754*** (4.23) | 0.0190*** (3.30) | -0.085*** (-8.17) | 0.007 (1.00) | 388.185* (1.73) | 0.019*** (2.94) |
| Tech-inv | | 3.72e-06 (0.08) | | | | |
| Tech-int | | | | -0.296*** (-5.67) | | |
| Highhum | | | | | | 3.46e-06 (1.07) |
| Constant | 628.391* (1.68) | 0.696*** (12.56) | 0.823*** (2.80) | 1.159*** (9.17) | 19 724.59*** (4.74) | 0.873*** (5.94) |
| Fixed effect | Yes | Yes | Yes | Yes | Yes | Yes |
| Observation | 90 | 90 | 90 | 90 | 90 | 90 |
| R-square | 0.408 | 0.535 | 0.717 | 0.529 | 0.570 | 0.299 |
| Bootstrap analysis 95% confidence interval | | [-0.046, -0.012] | | [-0.011,0.013] | | [-0.027,-0.001] |
| Total effect ($\alpha_1$) | 0.023 | | | | 0.023 | |
| Direct effect ($\gamma_1$) | 0.018 97 | | | | 0.019 0 | |
| Mediation effect ($\beta_1\gamma_2$) | 0.004 03 | | | | 0.004 | |
| Results | Partial mediation effect(17.5%) | | No mediation effect | | Partial mediation effect(17.4%) | |

Notes

\*\*\*, \*\*, and \* in the table denote significance levels at 1%, 5%, and 10%, respectively, and standard deviations are in parentheses, as below.

personnel, and with the increase in demand, the number of high-skilled personnel also increases, therefore, the development of the digital economy is conducive to the promotion of the advanced structure of human capital. In addition, the study found that the development of the digital economy in the Pearl River Delta is negatively correlated with the import of technology. On the one hand, because the import of technology not only requires great capital investment, but also inhibits the vigor of innovation, resulting in the negative impact of the import of technology is greater than the positive impact of the technological spillovers generated by the import of technology, on the other hand, with the development of the digital economy, the domestic information technology has been greatly improved, due to the technological barrier, the marginal utility of the import of technology has decreased substantially, and the marginal reward brought by independent innovation is much larger than the marginal reward brought by the import of technology. With the development of digital economy, the requirements for the level of digital technology are getting higher and higher, and the development of the domestic digital technology is getting more and more mature, and no longer relies on the import of technology. If the digital economy wants to realize the long-term and stable development, it is necessary to grasp the technology in its own hands.

This paper uses the three-step's the mediating effect model, introduces the level of independent innovation, the import of technology, and human capital as intermediary variables, and analyzes the path of digital economy affecting the upgrading of manufacturing structure in the Pearl River Delta. According to (9) in Table 3, after adding the level of independent innovation, the coefficient of the digital economy is 0.019, which is less than the result of (5) in Table 2, which is 0.023, and the result is significant. Furthermore, the coefficient of independent innovation level is positive, but the result is not significant. According to the test steps of the mediating effect model in Fig 2, the Bootstrap test was carried out, and the confidence

interval was [-0.046, -0.012], excluding zero, indicating that there was a partial mediating effect, and the mediation effect accounted for 17.5%. The level of independent innovation is the transmission path for the digital economy to promote the upgrading of manufacturing structure in the Pearl River Delta. According to (11) in Table 3, after the import of technology, the coefficient of the digital economy is 0.007, the result is less than 0.023, but the result is not significant. What's more, the coefficient of the import of technology is significantly negative, and bootstrap test shows that the confidence interval is [-0.011, 0.013], including zero, thereby hypothesis 2 is not entirely correct, independent innovation has intermediary effect, but the import of technology does not. The import of technology is not an intermediary variable for the digital economy to promote the upgrading of manufacturing structure in the Pearl River Delta. According to (13) in Table 3, when the intermediary variable of human capital structure upgrading is added, the coefficient of digital economy is 0.019, less than 0.023, and the result is significant. The result of human capital structure upgrading is positive, but not significant. Bootstrap analysis is conducted, and the confidence interval is [-0.027, -0.001], excluding zero, it indicates that there is a partial mediating effect, and the mediation effect is calculated to account for 17.4%, which confirms that the previous hypothesis 3 is correct, therefore, the digital economy can promote the upgrading of manufacturing structure in the Pearl River Delta region by promoting the upgrading of human capital structure.

## Robustness test

This article adopts the method of stepwise regression and sequentially adds variables for regression. The regression results are shown in Table 2 (1)—(5), and the coefficients and significance of the variables clearly indicate robustness. In addition, considering that the digital economy may have a certain time lag effect, this article uses the digital economy to replace the digital economy variable with a lag period for re regression. The results are shown in Table 2, Model (6), and the positive and negative directions and significance of the variable coefficients are completely the same, indicating robustness. Considering the measurement of the dependent variable, referring to Donghua Yu and Kun Zhang (2020) [32], the manufacturing industry is divided into labor intensive manufacturing, capital intensive manufacturing, and technology intensive manufacturing, among those, Labor intensive manufacturing industry including agricultural and sideline Food processing, food manufacturing industry, rubber and plastic products industry, metal products industry, textile industry, textile clothing and clothing industry, leather, fur, feather and its products and shoemaking industry, wood processing and wood, bamboo, rattan, palm, grass products industry, furniture manufacturing industry, non-metallic mineral products industry, printing and recording media reproduction industry, cultural and educational, arts and crafts, sports and entertainment products manufacturing industry. And the capital intensive manufacturing industry including petroleum processing, coking and nuclear fuel processing industry, Ferrous metal smelting and rolling processing industry, nonferrous metal smelting and rolling processing industry, chemical raw materials and chemical products manufacturing industry, chemical fiber manufacturing industry, general equipment manufacturing industry, wine, beverage and refined tea manufacturing industry, tobacco products industry, paper making and paper products industry. And the technology intensive manufacturing industry including specialized equipment manufacturing, automobile manufacturing, railway, shipbuilding, aerospace and other transportation equipment manufacturing, electrical machinery and equipment manufacturing, instrument and meter manufacturing, pharmaceutical manufacturing, computer, communication and other electronic equipment manufacturing. The index of the upgrading of manufacturing structure is re measured (Manuhigh) using the ratio of technology intensive manufacturing output

Table 4. The robustness test results of the mediating effect model.

| Variables | Manuhigh | Manuhigh | Manuhigh |
|---|---|---|---|
| Digital | 0.021** (2.51) | -0.000 6 (-0.09) | 0.019** (2.27) |
| Tech-inv | 0.000 031 (0.60) | | |
| Tech-int | | -0.360*** (-6.71) | |
| Highhum | | | 4.47e-06 (1.10) |
| Constant | 1.060*** (6.14) | 1.332*** (9.54) | 0.977*** (5.08) |
| Fixed effect | Yes | Yes | Yes |
| Observation | 90 | 90 | 90 |
| R-square | 0.451 | 0.656 | 0.457 |
| Bootstrap analysis 95% confidence interval | [-0.048,-0.014] | [-0.015, 0.011] | [-0.038, -0.005] |
| Total effect ($\alpha_1$) | 0.023 | | 0.023 |
| direct effect ($\gamma_1$) | 0.021 | | 0.019 |
| Mediation effect ($\beta_1\gamma_2$) | 0.002 | | 0.004 |
| Results | Partial mediating effect 9.6% | No Mediating effect | Partial mediating effect 17.8% |

Notes

***, **, and * in the table denote significance levels at 1%, 5%, and 10%, respectively, and standard deviations are in parentheses, as below.

value to total manufacturing output value, and a robustness test is conducted. The results are shown in Table 2, Model (7) that the model coefficients and significance are basically consistent with the benchmark regression results, and it confirms the robustness of the results once again.

Table 4 tests the robustness of the mediating effect model. It refers to the above practice, replaces the measurement indicators of the upgrading of manufacturing structure, and conducts regression and test again. The results are shown in Table 4, which is basically consistent with the results in Table 3. Therefore, the results are robust.

## Conclusion and political implication

With the advent of the information age, the integration of digital economy and manufacturing has become the main way to promote the upgrading of manufacturing structure. The digital economy through digital technology can improve resource allocation efficiency, save production costs, and optimize industrial structure. The development of digital economy has a significant role in promoting the upgrading of manufacturing structure in the Pearl River Delta, and this role has a certain mediation effect. The digital economy mainly promotes the upgrading of manufacturing structure in the Pearl River Delta by improving the level of independent innovation and promoting the upgrading of the human capital structure. The mediation effect of the level of independent innovation is greater than the mediation effect of the upgrading of the human capital structure. In addition, the development of the digital economy is conducive to the improvement of the level of independent innovation and the advanced structure of human capital, but the import of technology is negatively correlated with the digital economy. The import of technology is not a way for the digital economy to promote the upgrading of manufacturing structure, and the import of technology is not conducive to the upgrading of manufacturing structure in the Pearl River Delta. Furthermore, the research results also

indicate that the progress of the upgrading of manufacturing structure in the Pearl River Delta is still in a steady improvement stage, and the technology and resource allocation are still not fully mature, technology R & D, industrial integration still require government support and regulation, and government intervention has a significant positive impact on the upgrading of manufacturing structure in the Pearl River Delta. In addition, the level of marketization and urbanization is not conducive to the upgrading of manufacturing structure in the Pearl River Delta, further confirming the instability of the stage of the upgrading of manufacturing structure in the Pearl River Delta, which cannot do without government intervention.

Based on the empirical results of this study and a real socio-economic environment, this paper has the following policy implications. Firstly, the development of the digital economy has brought new opportunities to the development of the manufacturing industry, and the manufacturing industry want to achieve further development, it is necessary to undergo a transformation of the upgrading of manufacturing structure. Enterprises should strengthen the integration of the digital economy and the manufacturing industry by fully leveraging the high integration characteristics of the digital economy and infiltrating the digital economy into all aspects of the manufacturing process, thereby improving the efficiency of manufacturing resource allocation and achieving economies of scale, which is the key to achieving the upgrading of manufacturing structure for enterprises. Secondly, enterprise should strengthen patent certification and management and enhance awareness of intellectual property rights to provide a good and fair development platform for independent technological innovation. What's more, government should increase investment in independent innovation and encourage technological innovation, cross integration, and application to provide broad development channels for independent innovation. At the same time, government should emphasize the dominant position of technological innovation in enterprises, encourage enterprises to independently research and develop advanced technologies, and reduce dependence on the import of technology, thereby achieve a leap in technological level towards international standards. Enterprise should hold the initiative of technical progress in their own hands and promote the high-speed development of the digital economy through technical progress. Thirdly, government can cultivate professional high-tech innovation and R & D talents by promoting school enterprise cooperation to improve the digital literacy of human capital, and enterprise can improve the efficiency of "learning by doing" through providing vocational and technical training for enterprise talents to promote the advancement of human capital structure and provide talent reserves for the integration of the digital economy and manufacturing industry. Fourthly, government should steadfastly and continuously strengthen support for high-end manufacturing and leverage the advantages of the national system to increase investment in infrastructure construction for the integration of digital economy and manufacturing industry, focus on basic R & D, and use digital technology as a breakthrough point for the upgrading of manufacturing structure, thereby accelerating the process of the upgrading of manufacturing structure.

## Supporting information

**S1 Data.**
(PDF)

## Author Contributions

**Writing – original draft:** Ting Chen.

**Writing – review & editing:** Songlan Zhou.

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
