## [Decision Letter · Decision Letter 0]

30 May 2024

PONE-D-24-01707The impact of digital economy on the upgrading of manufacturing structurePLOS ONE

Dear Dr. Chen,

Thank you for submitting your manuscript to PLOS ONE. After careful consideration, we feel that it has merit but does not fully meet PLOS ONE’s publication criteria as it currently stands. Therefore, we invite you to submit a revised version of the manuscript that addresses the points raised during the review process.

Comments from the Editorial Office: We note that one or more reviewers has recommended that you cite specific previously published works. As always, we recommend that you please review and evaluate the requested works to determine whether they are relevant and should be cited. It is not a requirement to cite these works. We appreciate your attention to this request.

We look forward to receiving your revised manuscript.

Kind regards,

Annesha Sil, Ph.D.

Associate Editor

PLOS ONE

Journal Requirements:

 [This work was supported by the Natural Science Foundation of China: "Gap Measurement, Leading Mechanism, and Innovation Leap Research of New Science and Technology Revolution Pilot Technology" (No. 71974041)].  

Reviewers' comments:

Reviewer's Responses to Questions

**Comments to the Author**

1. Is the manuscript technically sound, and do the data support the conclusions?

Reviewer #1: Yes

Reviewer #2: Partly

2. Has the statistical analysis been performed appropriately and rigorously? 

Reviewer #1: Yes

Reviewer #2: I Don't Know

3. Have the authors made all data underlying the findings in their manuscript fully available?

Reviewer #1: Yes

Reviewer #2: No

4. Is the manuscript presented in an intelligible fashion and written in standard English?

Reviewer #1: Yes

Reviewer #2: Yes

5. Review Comments to the Author

Reviewer #1: The abstract should be strengthened due to the quantitative indicators obtained as a result of the research.

Additional justification of individual issues is required, which is reflected in the attached document.

Reviewer #2: The motivation for the research can be improved. At the moment, the problem to be solved is not really clear. The contribution to the field is limited. I recommend that the authors delve deeper into the data to uncover additional findings that can contribute to this area of research. It is also crucial for them to clearly differentiate their findings from previous research. Anyhow, even the available market tests with their experiences in the case industries properly verified preliminarily the results (by retroperspective approach, without longitudinal follow up studies, e.g. Robotic process automation deployments: a step-by-step method to investment appraisal, Business Process Management Journal, Vol. 29, No. 8, pp. 163-187.). Results are presented and counted accordingly. Scientific novelty and contribution to discipline could be highlighted more clearly. The relations between academia and companies is rooted in the cultural aspects of organizations, so please also refer to RPA Experiments in SMEs Through a Collaborative Network. In: Camarinha-Matos, L.M., Boucher, X., Ortiz, A. (eds) Collaborative Networks in

Digitalization and Society 5.0. PRO-VE 2023. IFIP Advances in Information and Communication

Technology, vol 688. Springer, Cham

6. PLOS authors have the option to publish the peer review history of their article (what does this mean?). If published, this will include your full peer review and any attached files.

Reviewer #1: No

Reviewer #2: No

---

## [Author Response · Author response to Decision Letter 0]

11 Jun 2024

Dear editor and reviewers,

Thanks for offering us an opportunity to improve the quality of our submitted manuscript (PONE-D-24-01707, The impact of digital economy on the upgrading of manufacturing structure).We appreciate very much the reviewers’ constructive and insightful comments. In this vision, we have addressed all of these comments. We hope the revised manuscript has now met the publication standard of your journal.

We highlighted all the revisions with Track Changes.

On the next pages, our point-to-point responses to the queries raised by the reviewers are listed.

Reviewer #1: The abstract should be strengthened due to the quantitative indicators obtained as a result of the research.Additional justification of individual issues is required, which is reflected in the attached document.

Response: Thanks to the reviewers' comments, the article has been revised according to the suggestions one by one, the relevant responses are as follows

Comment 1: Abstract, It is worth detailing with quantitative values according to the obtained results 

Response 1: According to the reviewers' comments, We have added the following details: Among them, the mediating effect is 17.5% for the level of independent innovation and 17.4% for the level of the advancement of the human capital structure. The results of the study also found that the upgrading of manufacturing structure cannot be separated from government support, and the influence of government support on t the upgrading of manufacturing structure reaches 44.9%, and government deployment and control is conducive to accelerating the process of advanced manufacturing structure.

Comment 2: Introduction, It is necessary to add a link to the information source from which the quantitative data was used, as well as to update the data for 2023 or 2022.

Response 2: Thank you to the reviewers for their comments, which have been updated to the latest data as follows: Compared to 2010, the proportion of added value of China's manufacturing industry to GDP dropped continuously from about 32.46% to 26.29% in 2020. Compared with the previous year, the proportion of added value of China's manufacturing industry to GDP increased slightly, accounting for 27.55% in 2021, and the growth rate of added value of China's manufacturing industry was about 18.83%.

Comment 3: Reference, Here and further - references to the used literature must be issued in accordance with the requirements.

Response 3: The formatting of references in the main text has been modified according to the style of the journal.

Comment 4: Literature Review, The literature review should be strengthened by analyzing the latest scientific publications in the field of digitization and IT. It is also worth investigating the impact of digitalization on various spheres and aspects of economic activity.For example: Impact of information and communications technology on the development and use of knowledge https://doi.org/10.1016/j.techfore.2023.122519 Opportunities and threats of digital transformation of business models in SMEs. doi:10.14254/2071-789X.2022/15-3/9

Response 4: Thank you for the reviewer's comments, considering that this paper wants to highlight the relationship between the digital economy and the structuring of the manufacturing industry, therefore the impact of digitization/information technology is not highlighted in the literature review of this paper, and in the previous article of my research (Ting Chen. Measurement of Digital Economy Development Level and Analysis of Influencing Factors in Guangdong Province [J]. Research on Science and Technology Innovation Development Strategy, 2023,7 (02): 40-48.), a more detailed study of the digital economy was made, therefore, I do not repeat the description here, thank you again for the reviewer's comments, and if the follow-up is still needed to be supplemented, I will be supplemented with the complete state.

Comment 5: Research Hypothesis, The choice of the proposed components is not justified.

Response 5: Thanks to the reviewer's comments, because this paper is based on the theory of endogenous economic growth, which selects the variables of technological progress (knowledge), capital and labor, the endogenous economic growth model believes that technological progress is conducive to the increase in the efficiency of labor, and add technology into the labor function in the productivity function. This paper is based on the endogenous economic growth theory of the hypothesis and believe that technological progress (knowledge) not only affects the efficiency of labor, but also has an impact on the efficiency of capital operations. Therefore, WE made changes in the model, add the technology into the function of labor and capital, resulting in the text of the proposed components.

Comment 6: Research Hypothesis, Needs more detailed justification

Response 6: Thanks to the reviewers' comments, the following changes have been made: According to equation (11) and the analysis above, in the context of the digital economy, gA and gk are always greater than 0, and α is greater than 0 and less than 1. Therefore, the left side of the equation is always greater than 0, as the growth rate of knowledge and capital continues to increase, the output growth rate also continues to increase.

Comment 7: Research Hypothesis, It is worth strengthening the rationale with reference to the results of scientific research in this area. Example: Knowledge Management and Economic Growth: The Assessment of Links and Determinants of Regulation DOI: 10.7206/cemj.2658-0845.52

Response 7: We thank the reviewers for comments, and according to the suggestions , the article has cited the relevant literature, and the relevant corrections are as follows: Olena Oliinyk (2021)[16] states that factors such as the ability of new technologies to work with people, the ability to innovate, and the means of communication become determinants of the efficiency of economic development, the shortage of skilled workers slows down the development of business and leads to additional costs for the development of human capital [17], information and communication technologies are able to drive economic growth[18], and increasing the efficiency of innovation is essential for creating competitive advantages [19]. Accelerating the development of the digital economy helps to promote technological progress and the accumulation of human capital, thereby assisting in the structural adjustment of the manufacturing industry, which is of great significance for promoting high-quality development of the manufacturing industry.

Reviewer #2: The motivation for the research can be improved. At the moment, the problem to be solved is not really clear. The contribution to the field is limited. I recommend that the authors delve deeper into the data to uncover additional findings that can contribute to this area of research. It is also crucial for them to clearly differentiate their findings from previous research. Anyhow, even the available market tests with their experiences in the case industries properly verified preliminarily the results (by retroperspective approach, without longitudinal follow up studies, e.g. Robotic process automation deployments: a step-by-step method to investment appraisal, Business Process Management Journal, Vol. 29, No. 8, pp. 163-187.). Results are presented and counted accordingly. Scientific novelty and contribution to discipline could be highlighted more clearly. The relations between academia and companies is rooted in the cultural aspects of organizations, so please also refer to RPA Experiments in SMEs Through a Collaborative Network. In: Camarinha-Matos, L.M., Boucher, X., Ortiz, A. (eds) Collaborative Networks in Digitalization and Society 5.0. PRO-VE 2023. IFIP Advances in Information and CommunicationTechnology, vol 688. Springer, Cham

Response :Thank you for the reviewer's comments, in the digital development of today, all walks of life are trying to break through the bottleneck of industry development through digitalization, the manufacturing industry is no exception, China's manufacturing industry is affected by a number of factors, the development of the current stagnation, this paper attempts to analyze the impact of the digital economy of China's Pearl River Delta (PRD) on the manufacturing industry structure of the seniority, to promote the integration of the digital economy and the manufacturing industry, and to solve the difficult problem of the China's Pearl River Delta (PRD) which manufacturing industry development is hindered through the digital economy. I also realize the many shortcomings of this paper, the research scope is small, and the innovativeness still needs to be strengthened. And, I would like to thank the reviewers for their comments once again, which provide a space for thinking about the research of this paper. Considering the scope of this paper and the research object, as well as the length of the study, I will carefully study the relevant research literature proposed by the reviewers in my subsequent research, and make further studies.

Best wishes

Ting Chen

---

## [Decision Letter · Decision Letter 1]

2 Jul 2024

The impact of digital economy on the upgrading of manufacturing structure

PONE-D-24-01707R1

Dear Dr.  Chen,

We’re pleased to inform you that your manuscript has been judged scientifically suitable for publication and will be formally accepted for publication once it meets all outstanding technical requirements.

Kind regards,

Yuantao Xie

Academic Editor

PLOS ONE

Additional Editor Comments (optional):

Reviewers' comments:

Reviewer's Responses to Questions

**Comments to the Author**

1. If the authors have adequately addressed your comments raised in a previous round of review and you feel that this manuscript is now acceptable for publication, you may indicate that here to bypass the “Comments to the Author” section, enter your conflict of interest statement in the “Confidential to Editor” section, and submit your "Accept" recommendation.

Reviewer #1: (No Response)

Reviewer #2: All comments have been addressed

2. Is the manuscript technically sound, and do the data support the conclusions?

Reviewer #1: Yes

Reviewer #2: Yes

3. Has the statistical analysis been performed appropriately and rigorously? 

Reviewer #1: Yes

Reviewer #2: Yes

4. Have the authors made all data underlying the findings in their manuscript fully available?

Reviewer #1: Yes

Reviewer #2: Yes

5. Is the manuscript presented in an intelligible fashion and written in standard English?

Reviewer #1: Yes

Reviewer #2: Yes

6. Review Comments to the Author

Reviewer #1: (No Response)

Reviewer #2: The story telling and introduction section lacks clarity regarding the limitations of existing literature. The research purpose, critique of literature, findings are not tightly coherent.

7. PLOS authors have the option to publish the peer review history of their article (what does this mean?). If published, this will include your full peer review and any attached files.

Reviewer #1: No

Reviewer #2: No

---

## [Editor Report · Acceptance letter]

17 Jul 2024

PONE-D-24-01707R1 

PLOS ONE

Dear Dr. Chen, 

I'm pleased to inform you that your manuscript has been deemed suitable for publication in PLOS ONE. Congratulations! Your manuscript is now being handed over to our production team.

Kind regards, 

on behalf of

Professor Yuantao Xie 

Academic Editor

PLOS ONE